# An Action Research Framework for Religion and the Stigma of Suicide

Curtis S. Lehmann [1,*], Carol A. Leung [2], Ivana Miller [1] and Samuel M. Girguis [3]

1 Department of Psychology, Azusa Pacific University, Azusa, CA 91702, USA
2 Department of Social Work, Azusa Pacific University, Azusa, CA 91702, USA; cleung@apu.edu
3 Department of Clinical Psychology, Azusa Pacific University, Azusa, CA 91702, USA; sgirguis@apu.edu
* Correspondence: clehmann@apu.edu

**Abstract:** Religious beliefs and practices have historically been intertwined with stigmatizing attitudes and responses to suicide, including stereotypes, prejudice, and discrimination. Understanding the relationship between religion and suicide stigma requires identifying specific religious beliefs and practices about suicide and how these are informed by broader worldviews, such as ethics, anthropology, and afterlife beliefs. Yet, research in this area has been complicated by the complex multidimensional nature of stigma and the diversity of religious beliefs and practices, even within religious traditions. Moreover, contrary arguments about the role of religious views of suicide in suicide prevention, specifically whether religious stigma is protective or instead contributes to risk, have obscured the interpretation of findings. This paper aims to advance research on this topic by first summarizing pertinent empirical findings and theoretical perspectives on public and personal stigma towards people with suicidal ideation (PWSI), people with suicidal behavior (PWSB), and suicide loss survivors (SLS). Secondly, a culturally nuanced action research framework (ARF) of religious stigma towards suicide is provided to guide future research. According to this ARF, research should advance strategically by investigating associations of religious beliefs and practices with stigmatization, identifying empowering resources within particular religious traditions, supporting suicide prevention efforts, and developing effective interventions to support PWSI, PWSB, and SLS. Moreover, such research efforts ought to equip religious leaders, and healthcare professionals working with religious individuals, to reduce stigma towards suicide and further the goal of suicide prevention.

**Keywords:** stigma; suicide; religion; spirituality; mental illness; suicide prevention





## 1. Introduction: Religions and the Stigma of Suicide

Durkheim (Durkheim 2005) inaugurated the field of suicidology by arguing religion provided social regulation with the potential to protect against suicide. As a result, most of the research on religion and suicide has focused on religion's protective role against suicide. Notably, a meta-analysis has shown religion protects individuals from suicide while noting some variation in certain cultures (Wu et al. 2015). Similarly, religiousness also seems to protect individuals from suicide attempts (Lawrence et al. 2016) in most major religious denominations (Gearing and Alonzo 2018; Stack and Kposowa 2011). In one recent study, there was a negative relationship between suicide ideation and attempts and dimensions of religion (e.g., moral objection of suicide), suggesting that religiousness plays a significant role in reducing suicide attempts and ideations (Jongkind et al. 2019).

Most have argued that the normative beliefs against suicide have aided suicide prevention (Carpiniello and Pinna 2017; Dervic et al. 2004). Durkheim considered it self-evident that the strict rules against suicide within Catholicism were a major contributing factor for lower suicide rates among Catholics than Protestants. In support, religious beliefs have been shown to be at the core of moral objections to suicide, including the beliefs that suicide is against the teachings of one's religion, only God has the right to end a life, or

that suicide is punishable to Hell (Linehan et al. 1983). These moral objections are related to the reduced risk of suicidal intent and attempts (Van den Brink et al. 2018). Similarly, religious importance has been shown to be strongly related to suicide acceptability (Stack and Kposowa 2011). On the other hand, a multinational study of 11 Muslim countries found that suicide acceptability partially mediated the relationship between religiousness and suicidality among those who had suicidal ideation and attempt (Eskin et al. 2020), indicating that religiousness decreased suicide risk through beliefs about the acceptability of suicide.

Although these research findings suggest that religious and moral views of suicide are protective against suicide, these beliefs have been noted to have negative effects on people affected by suicidality. Those who have attempted suicide have reported experiencing various types of stigma, including the perception of being "bad", which may decrease help-seeking behaviors (Rimkeviciene et al. 2015). Although the relationship between moral views of suicide and stigma is not entirely clear, scales that measure stigma towards suicide often include items inferring defective morality. For example, labeling people who die by suicide as "immoral" and "unforgivable" (Batterham et al. 2013) and blaming oneself for suicidal thoughts or behavior (Rimkeviciene et al. 2019) suggest low moral character. Thus, religious/moral views seem to contribute to stigma towards suicide.

Given the concurrent beneficial and detrimental effects, there is a need for expanded research on the role of religiousness on stigma towards suicide. Gearing and Alonzo (2018), in reviewing the theological and scientific findings on the relationship between the major religious traditions and suicide, argued strongly for expanded research on how religions might stigmatize suicide. This research is critical for understanding how religious communities can respond effectively to the problem of suicide, so that stigma is minimized while suicide is prevented. Faith communities are integral partners in suicide prevention (Suicide Prevention Resource Center 2021). As such, it is critical to understand the role of religion in the stigmatization of suicide.

*Scope of the Paper*

In this paper, we present a framework for engaging in research on the relationship between religion and the stigma of suicide. This action research framework (ARF) was designed to help researchers strategically develop studies that will address the existing gaps in this literature. We argue that stigma should be conceptualized using a social-psychological framework embedded within a sociological-ecological perspective that appreciates the broader systemic influences shaping stigmatizing beliefs, attitudes, and behaviors. Thus, we argue for a multidisciplinary action research agenda that addresses the psychosocial-spiritual interrelationships between religion and the stigma of suicide.

Although the concept of a research framework was loosely inspired by the "Research Agenda for Reducing the Stigma of Addictions" (Corrigan et al. 2017b, 2017c), the ARF was distinctly developed to encapsulate the broader and specific religious beliefs related to suicide. The ARF was developed through a series of technical discussions among the research team of two psychologists, a social worker, and a psychology graduate student. The first author began with a series of questions related to the relationship of religion to suicide stigma, which then was systematically grouped by the research team into the ARF. The group met regularly over six months to discuss and refine these concepts so that the ARF captured social-psychological, multidisciplinary, sociocultural, and ecological elements while emphasizing parsimony and flexibility.

The ultimate aim is to equip religious leaders and religious communities, along with healthcare professionals working with religious individuals, with strategies on reducing the stigma of suicide while maintaining an emphasis on suicide prevention. Thus, the ARF promotes strategic research to develop knowledge of concepts, relationships, and interventions to support religious communities. Although individual studies will likely involve basic research that may not provide direct application, the overall body of research on this topic should be translational for dissemination and implementation in religious

communities. Moreover, this research should be implemented to include people and communities who represent diverse religious traditions broadly. The benefits and burdens of this research are equitably and justly distributed (United States National Commission for the Protection of Human Subjects of Biomedical, & Behavioral Research 1978). Although effective research designs may require independently studying people from particular religious traditions, the overarching goal in this body of research would be to provide insights for people from religious traditions to destigmatize suicide.

The primary intended audience of this paper is researchers, given that it supplies a research framework. However, the paper also provides an overview of concepts relevant for clergy and faith leaders, and mental health professionals working with religious individuals to consider if they are engaging in suicide prevention or stigma reduction efforts.

## 2. Defining Stigma

The foremost concern in developing a research framework on religion and the stigma of suicide is to operationalize stigma properly. Unfortunately, stigma has sometimes been defined in vague and varied terms (Fox et al. 2018; Link and Phelan 2001; Pescosolido and Martin 2015). Link and Phelan (2001) tried to overcome this deficit and proposed a model of stigma that has been highly influential over the past two decades. They argued that stigma is a series of processes contingent upon the exercise of power. The first process is labeling, whereby some attributes are seen as a mark that distinguishes the person. The second process is setting apart, where the label is linked to negative attributes. The third process is the separation of "us" and "them" that robs people of their status, instead of defining people by their attributes or disease. The fourth process is discrimination, where the person experiences unfair behavior that disadvantages or harms them.

However, in recent years, there have been significant advancements in the conceptualization of stigma from both social-psychological and sociological perspectives. The social psychological perspective would define stigma as three separate components: stereotypes, prejudices, and discrimination (Corrigan and Kleinlein 2005; Corrigan and Kosyluk 2014; Fox et al. 2018). Stereotypes are the cognitive aspect of stigma and include beliefs about the traits and conduct of individuals. Prejudices are the affective component of stigma and involve emotional reactions to people within a specific class. Finally, discrimination is the behavioral component of stigma and involves unfair behaviors directed towards others. The term "stigma" thus comprises all of these concepts under a unified construct.

Moreover, stigma can be further divided into public and personal stigma. Public stigma involves the stereotypes, prejudices, and discriminatory behaviors present within the community. In this case, the public are the perpetrators of stigma (Phelan et al. 2008). Most research on stigma has focused on public stigma, with over 400 measures of mental illness stigma being developed (Fox et al. 2018). Yet, those researching mental illness stigma have emphasized the need to assess the experiences of the person being stigmatized (Livingston and Boyd 2010). An important element of this experience has been referred to as internalized stigma (Ritsher et al. 2003), self, and felt stigma (Livingston and Boyd 2010), and involves the internalization of negative stereotypes and perceptions. Yet, the experience can also include perceptions and experiences that have not been internalized (Rimkeviciene et al. 2019). Thus, we utilize the broader term personal stigma because it incorporates the perceptions of potential stigma, experiences of stigma, and self-stigma of those to whom the stigmatized label applies (Rimkeviciene et al. 2019).

This understanding of stigma can be furthered with sociological perspectives that consider how stigma exists within embedded systems that create and sustain these stereotypes, prejudices, and discriminatory actions. For example, Pescosolido and Martin (2015) argued the "stigma complex" involves individual-level components, such as disease characteristics, social characteristics, and behaviors, as well as community-level components, including the national context, media context, and social network context. These systemic community influences are important for understanding the impact of stigma and intervening appropriately to decrease stigma.

## 2.1. Stigma and Suicidality

The conceptions of stigma advanced above were general, reflecting the psychological and sociological states and processes through which stigma takes shape, regardless of the stigmatized attribute. Most of these conceptualizations were developed in research on mental illness stigma, as the stigma towards suicide has not been as thoroughly studied as mental illness stigma (Rimkeviciene et al. 2019). However, the stigma of suicidality has particular aspects, including unique stereotypes, prejudices, and discriminatory behaviors (Sheehan et al. 2017). Although there is an association between mental illness and suicidality, as many who have suicidal thoughts and behaviors may have a mental illness, the public and personal stigmas of suicide are distinct from the stigma of mental illness (Corrigan et al. 2017a; Rimkeviciene et al. 2019).

### 2.1.1. Public and Personal Stigma of Suicide

The public stigma of suicide, assessed as beliefs about those who have died by suicide, has been characterized with stereotypes of general negative attributes, isolation/depression, and glorification/normalization (Batterham et al. 2013). Other stereotypes of those who attempt suicide include the perception of the individual as weak, crazy, and distressed (Corrigan et al. 2017a), attention-seeking, selfish, incompetent, and immoral (Sheehan et al. 2017), and unintelligent, untrustworthy, and a personal failure (Scocco et al. 2012). The prejudices towards suicide have not been studied as extensively, but attitudes towards those who have attempted suicide have been found to load on two factors: fear/distrust and anger (Corrigan et al. 2017a). Finally, the discriminations towards those who have attempted suicide can include social distance and not accepting a person who has attempted suicide as a friend, teacher, employee, or dating partner (Scocco et al. 2012). Similarly, Corrigan et al. (2017a) have found that discriminations load onto avoidance, disdain, and coercion factors.

More recently, a measure of personal stigma has been developed and tested for validity using confirmatory factor analysis, showing that the scales predicted distress following a suicide attempt (Rimkeviciene et al. 2021). Two aspects of personal stigma, perceived and experienced stigma, are difficult to distinguish and seem to involve rejection and minimization, whereas self-stigma predominantly involves self-blame (Rimkeviciene et al. 2019, 2021). The research on personal stigma, though just emerging, has emphasized that personal stigma has distinct processes that are unique from public stigma. For instance, perceptions that others will react negatively to those with suicidal history seemed to load separately according to social groupings of friends and family, work colleagues, and healthcare workers, rather than according to particular cognitive, emotional, or behavioral responses (Mayer et al. 2020).

### 2.1.2. Stigma towards Suicide Loss Survivors

After the death of a loved one, suicide loss survivors may experience bereavement differently than those that die of natural death (Hanschmidt et al. 2016). One factor that further impedes the grieving process among suicide loss survivors is the stigma (Sheehan et al. 2018; Sudak et al. 2008). Researchers have found two interacting dimensions of the stigma of suicide loss: public and internalized stigma (Hanschmidt et al. 2016). Public stigma can include being blamed or judged for the death by suicide (Evans and Abrahamson 2020; Sveen and Walby 2008). After a suicide, families reported stigma-related experiences, including strained communications with extended family and their community and unhelpful advice (Feigelman et al. 2009). Additionally, suicide loss survivors may be labeled with undesirable characteristics, and others may devalue, reject, or exclude them from social interactions. The stigma could depend on the person's relationship with the deceased. Research has found that the stigma of having a family member with mental illness varied depending on the person's role in the family (Corrigan et al. 2006).

Challenges also arise because talking about death, let alone suicide, may be uncomfortable for many to discuss (Feigelman et al. 2009; Goulah-Pabst 2021). Sheehan and

colleagues (2016) found distinct themes for public stigma among suicide loss survivors. The three most common themes shared with these researchers among suicide loss survivors include stereotypes, prejudice, and discrimination. From these themes, suicide loss survivors in the focus groups discussed a public perception of blaming and failure to help the suicide decedent from dying. Thus, people often refrain from seeking the support they may need for fear of having negative labels associated with them, including discriminatory reactions such as shunning, shushing, shaming, impatience, mistrust, denial of support (Sheehan et al. 2018), and threats to social bonds (Goulah-Pabst 2021).

Furthermore, Sheehan and colleagues found that internalized stigma, often discussed as self-stigma, was commonly present among suicide loss survivors. These individuals may internalize the public stigma and thus perceive the actions of others as negative judgment or rejection (Dunn and Morrish-Vidners 1988; Feigelman et al. 2009). In the focus groups, families experienced shame and often struggled with internalized thoughts of being "contaminated by suicide". As a result, suicide loss survivors may rather keep silent about the suicide. The secrecy about suicide loss or lack of disclosure to others about suicide could contribute to self-stigma (Oexle et al. 2020). While keeping silent about the suicide loss can avoid public stigma among suicide loss survivors, the internalizing thoughts and feelings about the stressful life event could increase emotional distress, causing social isolation and avoidance of social situations. In addition, the concealment and secrecy of the suicide from others may perpetuate stigma and result in negative outcomes, especially among children bereaved. As an example, the child may be denied the proper grieving process of a parent who died of suicide (Peters et al. 2016).

2.1.3. The Severity of Suicide Stigma

It is important to highlight the stigma towards suicide is more severe than stigma for depression (Sheehan et al. 2017. )There may be several reasons why suicide is uniquely stigmatized. First, suicide represents a violation of the instinct of self-preservation (Joiner 2010), which might be uniquely distressing to others. Alternatively, according to attribution theory, persons with conditions thought to be under the person's control are seen as more responsible and blameworthy (Corrigan et al. 2003). The stigma associated with suicide seems to imply those who attempt are able to control their actions, which may explain the severity of the stigma. In a study on non-suicidal self-injury, attributions of personal responsibility have been found to affect emotional responses, which can lead to changes in helping behaviors (Nielsen and Townsend 2018). Another explanation for stigma against suicide would be that suicide may be associated with notions of impurity that leads to a sense of disgust (Rottman et al. 2014). Some have argued that focusing less on purity and more on the harm of suicide might actually decrease the stigma (Mason et al. 2021). In any case, views of suicide are embedded in the combination of culture and the structure of society to multiply or attenuate the consequences of such stereotypes, prejudices, and discrimination (Pescosolido and Martin 2015).

There is some indication that stigma towards suicide has decreased in recent years (Witte et al. 2010). However, research on stigma has a history of making claims of decreasing stigma that turned out to be incorrect (Pescosolido 2013). Given the long history of suicide stigma, it seems reasonable to anticipate that stigma will remain an intractable problem for most modern societies.

**3. A Working Model of the Stigma of Suicide**

Figure 1 depicts the social-psychological understanding of the stigma of suicide, including public and personal stigma, as it relates to people with suicidal ideation, people with suicidal behavior, and suicide loss survivors. A further step in the conceptualization of how public stigma is formulated as stereotypes, prejudice, and discrimination, personal stigma is formulated as perceived stigma, experienced stigma, and self-stigma. This model is conceptually similar to one developed by Fox et al. (2018), which addressed mental illness stigma. Although the sociological aspects of stigma were not included in Figure 1,

these systemic and society-level facets are crucial to understanding the public and personal stigma of suicide.

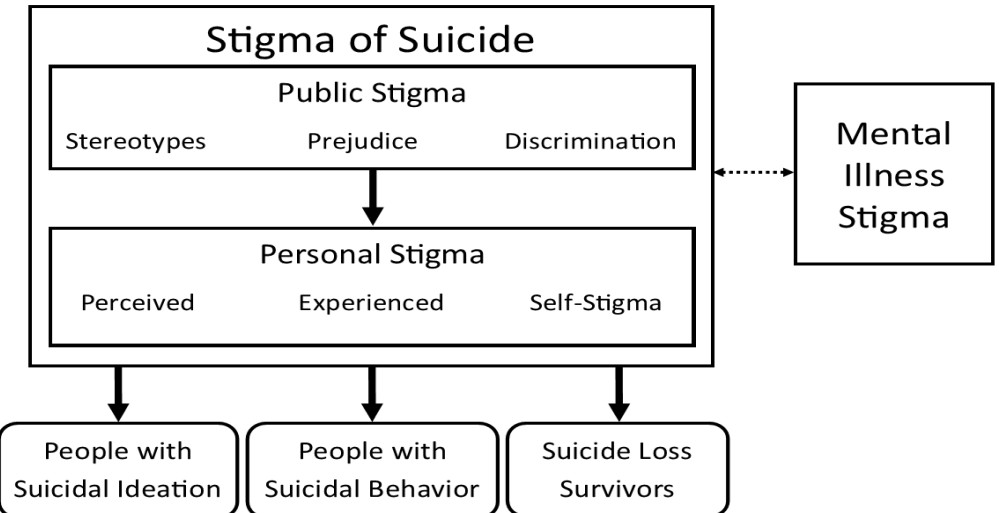

**Figure 1.** Model of public and personal stigma of suicide.

*3.1. Effects of the Stigma of Suicide*

3.1.1. Effects of Stigma on People with Suicidal Ideation or Behavior

Regardless of the underlying causes of stigma, the research indicates stigma towards suicide has negative effects. The public stigma of suicide is thought to cause social distancing, or avoidance, towards those who attempt suicide (Corrigan et al. 2017a; Lester 1993). As a result, public stigma can be a barrier to gatekeeping (Burnette et al. 2015), including difficulties with saying the word "suicide" (Monteith et al. 2020). The (World Health Organization (n.d.) has recognized that one myth of suicide is that "someone who is suicidal is determined to die". This stereotype of PWSI and PWSB could discourage attempts to intervene or refer people at risk of suicide.

The public stigma towards suicide can, in turn, be perceived or directly experienced by those with lived experience (Rimkeviciene et al. 2019). The perceived stigma by those with lived experience seems to cause many PWSI and PWSA to withhold their history of suicidality to their support system and health care providers (Sheehan et al. 2019). One type of perceived stigma, termed anticipated stigma, has been found to be associated with increased risk of suicide among suicide attempt survivors, mediated by the belief they must keep their suicidal history secret (Mayer et al. 2020). This perceived need to conceal their suicidal history from loved ones can produce feelings of isolation, whereas disclosure can result in relief from depression symptoms (Frey et al. 2016). The fear associated with disclosure can also be a barrier to seeking out services when needed (Arria et al. 2011; Calear et al. 2014; Chan et al. 2014).

If public stigma is internalized and accepted by PWSI or PWSB, this can result in self-stigma. Although the effects of the self-stigma of suicidality have not been thoroughly studied, research on the self-stigma of mental illness suggests that it could lead to increased depression, decreased self-esteem, and greater symptomatology (Boyd et al. 2014). Moreover, self-stigma can exacerbate distress and risk for suicidality for PWSI and PWSB. Lack of disclosure of suicidal history due to self-stigma can lead to feelings of shame or stress from keeping a secret (Sheehan et al. 2018). The self-stigma of mental illness is associated with increased suicidal thoughts (Xu et al. 2016) including at 2-year follow-up, even after controlling for baseline suicidality (Oexle et al. 2017). Together, these effects can potentially put a person at further risk of suicide, noted as the reciprocal effect of stigma and suicide (Carpiniello and Pinna 2017; Pompili et al. 2003).

### 3.1.2. Effects of Stigma on Suicide Loss Survivors

The stigma for surviving a suicide is greater than for surviving a natural death (Hanschmidt et al. 2016) and can have adverse consequences on suicide loss survivors (SLS; Doka 2002, p. 327; Rando 1988, pp. 111–14). There is some indication the levels of distress among suicide loss survivors were positively associated with the level of perceived negative public attitudes towards suicide (Scocco et al. 2017). SLS often feel awkward and uncomfortable about the suicide of their loved one, which leads to avoidance and secrecy (Evans and Abrahamson 2020). Perhaps, for this reason, SLS are at higher risk of complicated bereavement (Tal Young et al. 2012).

Public stigma and self-stigma contribute to worse mental health outcomes to SLS (Oexle et al. 2020). Notably, stigma towards SLS is associated with suicidal ideation (Lester and Walker 2006; Hanschmidt et al. 2016; Pitman et al. 2014) among suicide loss survivors. The perception of higher levels of stigma has been found to be associated with overall depression, self-harm, and suicidality (Evans and Abrahamson 2020). Losing a child to suicide has been found to be associated with depression and suicidal thoughts (Feigelman et al. 2009).

Pitman and colleagues argued suicide loss survivors who are exposed to suicide of a close contact may find themselves with negative health and social outcomes, depending on the relationship to the suicide decedent. Specifically, there is an increased risk of suicide among those bereaved by suicide, increased risk of psychiatric care admission for parents bereaved by the suicide of an offspring, increased risk of suicide in mothers bereaved by an adult child's suicide, and increased risk of depression (Pitman et al. 2014).

### 3.2. Differentiating Subpopulations Affected by the Stigma of Suicide

When researching stigma towards suicide, it is critical is to recognize the distinction between PWSI and PWSB. The unique experiences of these two related but distinct populations is sometimes overlooked by referring to those with these experiences as "suicidal." Arguably, there are times when grouping these populations together may be helpful for communication, so the broader concept of "suicidality" is sometimes employed in this paper. Yet, the general research literature on suicide has been emphasizing the unique characteristics of these populations. This distinction was prominently highlighted in Thomas Joiner's (2005) "Interpersonal-Psychological Theory of Suicide" (IPTS), wherein he posited independent risk factors for suicidal ideation and suicide attempt. Since IPTS was developed, there have been several additional theoretical approaches that have reiterated this distinction between suicidal desire and suicidal capability (Galynker 2017; Klonsky and May 2015; O'Connor and Kirtley 2018). This overall approach has been termed the ideation-to-action framework (Klonsky and May 2014; Klonsky et al. 2018).

The second reason for distinguishing PWSI and PWSB is the distinct types of stigma that each of these populations might face (Corrigan et al. 2017a). Although research on the stigma of suicide has not identified clearly distinct stigma for PWSI and PWSB, the measures involved in studying this topic are often specific to one group. For instance, the Stigma of Suicide Scale (SOSS; Batterham et al. 2013) addresses stereotypes of suicide decedents whereas the Stigma Towards Suicide Attempters scale (STOSA; Scocco et al. 2012) assesses its eponymous population. The Personal Suicide Stigma Questionnaire asks people to report on stigmatization related to "suicidal thoughts or behavior" that unfortunately fails to distinguish potential unique aspects of stigma towards these two groups. Despite the lack of research on whether PWSI and PWSB are subjected to qualitatively or quantitatively different stigma, the ARF presumes that differences are present and thus argues for differentiation.

The third population included within this ARF is suicide loss survivors. Although suicide is a relatively rare event, it is estimated that for every suicide decedent, there are an average of six survivors who are severely bereaved by their death (Berman 2011; Pompili et al. 2013; Shneidman 1969). Over time, the number of SLS increases as thousands of individuals are affected by a suicide death each year (Crosby and Sacks 2002). SLS have

different experiences from other types of death loss given that the survivor's association with the suicide decedent can lead to stigma that is manifested in how they are treated by others who are aware of their loss. Because of the stigma associated with suicide, SLS often experience concealment of the suicide death, social withdrawal, impaired psychological and physical health, and complicated bereavement (Hanschmidt et al. 2016). Additionally, SLS are more likely to disconnect with their social networks after their deaths compared to those bereaved by accidents and natural deaths. Thus, peer-facilitated suicide survivor support group meetings may help this population empower each other through the healing process (Feigelman and Feigelman 2008).

In this paper, we have chosen to use language that we believe is most likely to protect against the labeling and stereotyping process. The research literature has occasionally referred to people with suicidal ideation as "ideators" and people with suicidal behavior as "attempters", (see Klonsky and May 2015, others). Although this terminology was likely utilized as a shorthand to allow for clear communication, it risks labeling a person according to their suicidal experiences and losing their other distinctive identities. For this reason, we have chosen to use person-first terminology in this paper, referring to people with suicidal ideation (PWSI) and people with suicidal behavior (PWSB). To describe those who have lost a loved one to suicide, we utilize the phrase suicide loss survivors (SLS) because of the positive connotations associated with being a survivor and the linguistic challenges of using person-first language. The phrases "people who have lost someone to suicide" or "people who have survived suicide loss" were seen as being too wordy and cumbersome.

### 3.3. Populations Excluded from This Action Research Framework

Arguably, this ARF could also include multiple other populations as there are numerous other types of deaths associated with suicide that are excluded, including martyrs, suicide bombers, mass suicides, perpetrators of murder-suicide, and perpetrators of mass killings terminated by suicide. These groups have been excluded for the sake of simplicity.

There are three other populations excluded from the ARF that we wish to comment upon. First, those who have died by suicide could be argued to be crucial for understanding the stigma of suicide. Certainly, stereotypes and prejudices can be held towards a suicide decedent. The Stigma of Suicide Scale (SOSS; Batterham et al. 2013) addressed stereotyped beliefs about people who have died by suicide. However, it is unclear whether a deceased individual can be stigmatized. According to Link and Phelan (2001), stigma results in discrimination, or mistreatment, of the individual. Indeed, there is certainly a long history of desecration of the corpses of suicide decedents (Barbagli 2015; Maris 2019; Weaver and Wright 2009). Nevertheless, for this ARF, we excluded this population to preserve a narrower focus and because desecrating practices are no longer evident in nearly all contemporary cultures. Instead, we argue that negative beliefs and emotional reactions towards suicidal decedents are shown in forms of stigma against PWSI, PWSA, and SLS.

Second, we also exclude physician aid in dying, often referred to as "assisted suicide." The American Association of American Association of Suicidology (2017) has drafted a position that physician aid in dying was distinct from suicide due to the context of this practice within the medical field, including legal requirements for a second opinion, provision of treatment alternatives, 15-day waiting periods, ruling out of mental illness that would impair decision making, and the bereavement being less severe for families. The authors do not make any claims of support or opposition to this position but choose to remain focused on conventional suicide.

Finally, those who engage in non-suicidal self-injury (NSSI) are not included in this review. Although there is evidence of a strong relationship between NSSI and suicidal behavior (Ribeiro et al. 2016), self-injurious behaviors are, by definition, carried out without suicidal intent. Although there may be overlapping aspects of the stigma of self-injury with suicidal behaviors, such as the presumption of suicidal intent, hiding of scars, and

labeling as attention-seeking (Staniland et al. 2021), the relationship between these stigmas and the stigma of suicide remains unclear.

## 4. Religious Views on Suicide

Research has documented the varied beliefs about suicide present within religious traditions and contemporary manifestations of these faiths. Not surprisingly, the specifics of these beliefs vary depending on the religion. In Judaism, the doctrine does not allow people to hurt or harm themselves (Bailey and Stein 1995; Gearing and Alonzo 2018; Schwartz and Kaplan 1992) because it goes against preserving human life (Nelson et al. 2012) and is seen as worse than murder, given that repentance cannot occur (Kaplan and Schoeneberg 1988). Similarly, Islam forbids suicide, with the Qur'an referring to suicide as self-murder (Colucci and Martin 2008; Nelson et al. 2012), with even the desire for death being forbidden (Shah and Chandia 2010). There is debate in the literature about whether Islam teaches that suicide leads to condemnation in Hell (Nelson et al. 2012), but many Muslims believe that Hell is the consequence of suicidal behaviors (Abou-Allaban 2004; Nelson et al. 2012).

Christianity can be roughly divided into Catholic, Protestant, and Orthodox traditions (Gearing and Alonzo 2018). The Catholic Church's Catechism has taught that the commandment, "Thou shall not kill", applies to the act of suicide (Nelson et al. 2012; Catholic Church 2019, para. 2325). Consequently, suicide decedents were historically barred from burial in a Catholic cemetery (Gearing and Lizardi 2009). Currently, the common practice is to pray for forgiveness of the deceased and comfort for those bereaved (Gearing and Lizardi 2009). This shift has been accompanied by an argument that those who died by suicide may have diminished culpability if the act of suicide was caused by "[g]rave psychological disturbances, anguish, or grave fear of hardship, suffering, or torture" (Catholic Church 2019, para. 2282). Protestants also see suicide as sinful but do not severely prohibit suicide (Teo et al. 2021). Views of suicide within Christianity since the Enlightenment have shifted to become more compassionate, emphasizing the love of neighbor and prevention of suicide (Nelson et al. 2012; Teo et al. 2021). There is limited research on the stigma of suicide in Orthodox Christian traditions; the research that has been conducted suggests suicide is perceived as less acceptable among Orthodox Christians (Eskin et al. 2019).

Buddhism is generally considered a religion, although many argue that it might be better classified as a philosophy (Teo et al. 2021). Buddhism condemns suicide because it is an expression of tanha, which is unenlightened worldly desire, and moha, which is delusion in that the person wrongly believes suicide will solve their problems (Teo et al. 2021). Buddhism recognizes suicide as a means to end suffering but upholds that it does not achieve this aim, with those who die by suicide being unprepared for the next life (Gearing and Alonzo 2018; Gearing and Lizardi 2009). Suicide would result in the person being reborn into a lower hierarchy of existence. In other words, after reincarnation of being reborn, the individual would experience additional suffering (Teo et al. 2021).

Hinduism is a multifaceted religion with diverse belief systems (Teo et al. 2021). Hinduism generally views suicide negatively as an act against the good of humanity (Lakhan 2008). The concept of Dharma conveys a duty to family, society, and the universe that may protect against suicide (Nelson et al. 2012). However, Hindu philosophy believes that death leads to rebirth through reincarnation, which may lead to more permissive views towards suicide (Gearing and Alonzo 2018), though suicide could be argued to be a "bad death" where the person is reincarnated to a lower level of existence where they will experience suffering or have an animal life (Leach 2006).

The above research identified the beliefs of various religious traditions on the act of suicide. However, it is imperative to note that religion cannot be fully understood without exploring the culture. The relationship between religion and culture is complex but the two are inextricably linked (Saroglou and Cohen 2011; Alothman and Fogarty 2020). Culture and religion impact societal and individual views of suicide. For example, Cleary and Brannick (2007) explored how views of suicide have changed in Ireland. They noted that

even though the country has experienced increased secularization, many in Ireland still hold traditional views based on religious beliefs. However, these beliefs differ greatly between more rural areas of the country and more urban areas of the country (Cleary and Brannick 2007). Stack (1998) analyzed the view of suicide among African Americans and noted a complex interrelationship between gender, culture, religion, and other situational differences. Stack argued for a more developed understanding between the variables that impact suicide in specific populations.

## 5. The Relationship between Religion, Suicide, and Stigma

The role of religion in stigma is an area needing further research as current measures of suicide stigma lack religious factors (Ghasemi et al. 2015). In particular, religion's contribution to suicide stigma is under-researched (Moksony and Hegedűs 2021). This is particularly troublesome given research and vital statistics demonstrating religious differences in suicide mortality outcomes (Moksony and Hegedűs 2019). For example, suicide attempts were more common among depressed patients with a religious affiliation than those who were religiously unaffiliated, and suicidal ideation was associated with the increased importance of religion (Lawrence et al. 2016). Although not explicitly studying stigma, Linehan et al. (1983) found moral objections to suicide, entailing religious proscriptions against suicide provided a "reason for living" that could curtail suicidal behavior. More recent research has confirmed that moral objections are independently associated with decreased suicide attempts, even when controlling for religious affiliation and other relevant demographic and clinical variables (Lizardi et al. 2008). In further support of this idea, the empirical literature has found moral objections to suicide to be associated with decreased suicidality (Van den Brink et al. 2018).

However, the literature on the topic has generally left uncontrolled other associated factors, such as religious support, religious coping, and religious identity, that could play a role in reducing suicidal behaviors (see Eskin et al. 2020 for an exception). Moral objections may be correlated with these other religious variables that could be more proximally related to suicide protection. For this reason, it is unclear whether religious proscriptions against suicide are the predominant factor in religion's protective effects against suicidal ideation, attempt, and death.

Moreover, the research on religion and suicide has not thoroughly addressed how religious beliefs and practices might promote norms against suicide in ways that may have deleterious effects on some individuals. One notable variable for the stigma of suicide is negative religious coping which reflects struggle in one's relationship with God, including beliefs of being punished or abandoned (Pargament et al. 2011). Negative religious coping has been found to be strongly associated with suicidal ideation (Currier et al. 2017; Trevino et al. 2014) and with suicide attempts (Eskin et al. 2020). This research indicates that negative religious coping, or spiritual struggle, is a risk factor for suicide. Rather than alleviating suicidal thoughts and behaviors, moral condemnation of suicide that demands divine punishment might contribute to suicidal thoughts and behavior.

One potential concern extending from these findings is whether to conceptualize the belief that God will punish those who die by suicide as a form of stigma. Arguably, negative religious coping could potentially be a form of personal stigma, specifically perceived stigma, if the beliefs about God's punishment are related to the person's suicidality. It is plausible that the belief of God forbidding suicide might lead to struggle when one experiences suicidal ideation or behaviors, as the person might believe that God is liable to punish them for these thoughts and actions.

However, characterizing negative religious coping as a stigma leads to a potential concern that science might be encroaching upon the truth claims of religious and spiritual views. An argument could be made that religious views of suicide are theological claims and should not be characterized as stigma. However, our position is that religious and spiritual views should not be excluded from scientific criticism, although some distinctions need to be made. In particular, stigma is understood as a process that labels, stereotypes,

sets apart, takes away status, and discriminates against *people* (Link and Phelan 2001, emphasis added), not the behaviors that people engage in. The behaviors can be involved in attributing a label but ultimately the label, and the consequent stereotypes, prejudices, and discriminatory behaviors are attached to a person. In other words, suicidal ideations and attempts are not subject to stigmatization but, instead, the person who experiences such thoughts and behaviors could be.

Consequently, religious views about suicidal *behaviors* would not be necessarily stigmatizing unless the beliefs suggested that the person would be labeled according to their behavior. For instance, the belief that suicide is a sin would not be stigmatizing if treated as one type of sin among many. However, arguing that suicide is an unforgivable sin or that it necessitated abandonment of faith would be understood to be stigmatizing because the person's identity and afterlife would be determined solely by their behavior. At the same time, researchers ought to appreciate that scientific perspectives are descriptive of observable reality and may not reflect the complete ontology of religious individuals.

### 6. Action Research Framework for Religion and the Stigma of Suicide

In Figure 2, we propose an action research framework for the empirical and theoretical study of religion and the stigma of suicide. The central construct in the ARF is "Responses to Suicide," which entails stigmatization and empowerment. The stigmatization of suicide was depicted in detail in Figure 1. The ARF indicates that responses to suicidality are the result of suicide-specific religious beliefs and practices. These suicide-specific religious beliefs and practices result from broader religious worldviews, including but not limited to ethics, anthropological beliefs, and afterlife beliefs. The ARF affirms that culture is thoroughly embedded in religious beliefs and practices, such that the two cannot be parsed (Saroglou and Cohen 2011; Alothman and Fogarty 2020). This ARF would uphold that research ought to study these constructs individually and in relation to responses to suicide. Additionally, further research could identify religious beliefs that are relevant to either stigmatization or empowerment of people affected by suicidality or suicide loss and that can be minimized, disputed, or replaced by alternative beliefs.

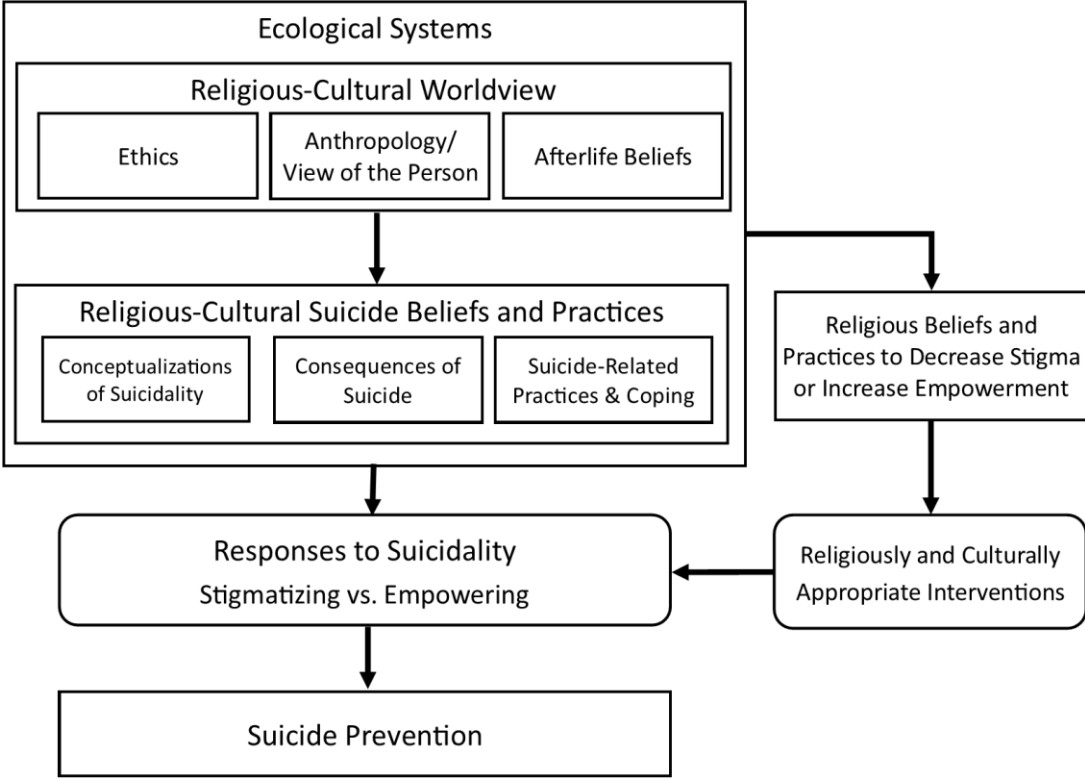

**Figure 2.** Action research framework for religious stigma of suicide.

Phrasing this as an *action* research framework affirms that research should progress with a commitment towards the reduction of the stigma of suicide, concurrent with the goal of suicide prevention. This framework would support the advancement of research studies with consideration to establishing a domain of knowledge and identification of interventions for decreasing stigma towards suicide and increasing empowering responses for those experiencing suicidal thoughts and behaviors, along with survivors of suicide loss. The ARF would also involve a strategic attempt to help people within religious communities, including faith leaders and congregants, and the health professionals working with those in the religious community. Such an approach would require an emic understanding of religious beliefs and practices, as universal conceptualizations would likely be less helpful for lessening the burden of stigma towards suicide.

As a clarification, this ARF would not require that each study on this topic be an applied research study with clear intent to lessen stigmatizing responses. Nor would the framework require a particular research methodology, such as participatory action research (Baum et al. 2006), though this methodology would certainly be encouraged. Rather, the continuum of research approaches, basic through applied, would be strategically developed with the intent to contribute to a literature that could be utilized for lessening stigma and affirming those who experience suicidal thoughts and behaviors. Empirical studies could involve quantitative, qualitative, or mixed methods analyses to contribute to the literature. Moreover, as mentioned above, the research could be multidisciplinary, such that philosophical, theological, psychological, and sociological perspectives would be valued as contributing helpful perspectives.

The outcome of such a research program would be twofold: (1) the development of a knowledge domain on various religious-cultural beliefs and practices related to suicide, including the associations with stigmatizing or empowering responses and (2) the identification of specific interventions that can be utilized by faith communities, within particular cultural contexts, to reduce the stigma of suicide and support the aims of suicide prevention.

### 6.1. Religious Conceptualizations, Consequence Beliefs, and Practices Related to Suicidality

The action research framework posits that religious-cultural views of suicide are the proximal cause of stigmatizing and empowering responses to suicide. These views of suicide could include religious conceptualizations, beliefs about religious consequences, and religious practices. To understand how religiousness is related to stigma towards suicide, researchers ought to establish a comprehensive understanding of the beliefs and practices observed in diverse religious communities. Although sacred texts and official religious sources may be useful in this venture, the actual perspectives of religious individuals should be surveyed, including their degree of acceptance of "official" positions, as well as any alternative beliefs and practices that are prevalent.

6.1.1. Religious Conceptualizations of Suicide
Typology of Suicidal Ideation and Behaviors

Suicide is typically assumed to be a unitary construct. Although Durkheim initially proposed a typology of suicide deaths based upon the influence of social regulation and social integration processes, typologies of suicide have not been commonly employed in contemporary scholarship (for an exception, see Pridmore and McArthur 2010). One scientific definition of suicide is given as "death caused by injuring oneself with any intent to die" (Centers for Disease Control and Prevention 2018). Similar definitions exist for suicide attempt and suicidal ideation, indicating these behaviors are seen as homogenous within the scientific community.

Rather than seeing all forms of suicidality, including death, attempt, and ideation, as being represented by a single construct, religious views may help differentiate and distinguish these behaviors into different categories (e.g., sin vs. suffering). These categorical distinctions may have implications for stigmatization or empowerment. Religious beliefs typically influence conceptualizations of behaviors, including death (Garces-Foley

2014), suicide (Gearing and Alonzo 2018), and stigma (Larkin 2020). As such, these beliefs may provide a typology or hierarchy of various expressions of SI and SB distinct from scientific and clinical perspectives. Meaningful and consequential conceptualizations of various forms of suicide, suicide attempt, and suicidal ideation likely exist within religious traditions. Typologies could also distinguish between suicidal ideations, such as viewing passive suicidal ideation (i.e., wish for death) as qualitatively different from active suicidal ideation (e.g., actual thoughts of killing oneself), rather than only viewing differences quantitatively as some scientific conceptualizations seem to imply. These distinctions may even extend to tertiary issues of suicide such as martyrdom, physician-assisted death, jihad, or mass suicide in religious cults.

Based on affiliation and traditions, these understandings of suicidal ideations and behaviors likely vary between and among religiously affiliated populations and subpopulations. Differentiations are likely expounded upon by conceptualizations of sin, morality, justice, charity, and other themes and principles guiding religiously motivated behaviors and practices. Additionally, ethnocultural differences in conceptualizations of death, suicide, and stigma may also interact with religious conceptualizations and yield different secondary and tertiary typologies and hierarchies. Developing an increased understanding of differentiations in conceptualizations of death, suicide, and stigma, with consideration of religion, culture, and other diversity variables, may aid researchers and clinicians in suicide theory construction and application as well as in the development of predictive risk algorithms.

There is potential for misunderstanding if religious distinctions within SI and SB are not appreciated and understood both within and outside of religious communities. Droge and Tabor (1992) suggested many early church Christian martyrs could be characterized as suicides, as many sought out their deaths, but this characterization has been questioned more recently (Buck 2012) and conflicts with the traditional, emic view. Similarly, framing those who kill themselves in the midst of a religiously motivated attack of violence as "suicide terrorists" does not reflect the view that these actions are perceived as distinct from suicide. Catholic teachings seem to suggest a typology where suicides that are "committed with full knowledge and deliberate consent" (Catholic Church 2019, para. 1857) are mortal sins but those caused by "Grave psychological disturbances, anguish, or grave fear of hardship, suffering, or torture" (Catholic Church 2019, para. 2282) may be characterized with less responsibility. Across religions, certain types of self-inflicted or willful death can be seen as acceptable or honorable in particular situations, such as when enacted for altruistic reasons or as a form of protest (Teo et al. 2021).

In regards to the stated purpose of the ARF, knowledge of religious typologies of suicide might be critical for the development of interventions for reducing the religious stigma of suicide. Interventions could utilize typologies with existing distinctions if particular types of suicide are less stigmatized. For instance, interventions could target Catholic views of suicide by demonstrating the vast majority of suicide deaths are caused by "grave psychological disturbances," as noted above. Another approach would be to make slight alterations to typologies that would be religiously and culturally appropriate for the context of the intervention. This could be achieved by emphasizing other religious teachings that are empowering in the views of people with SI and SB.

Key Takeaway for Researchers: Religious individuals may conceptualize suicidality in a manner distinct from scientific understandings. How these conceptualizations are applied to circumstances involving suicidality may be malleable, allowing for a potential target of intervention.

Etiology of Suicidal Ideation and Behaviors

Another factor for how religious beliefs can shape conceptualizations of suicide is by attributing suicidal thoughts and behaviors to particular causes. These attributions could be situational or dispositional, which could, in turn, affect the degree of stigmatization. For instance, if death by suicide is attributed to the dispositional trait of weak (or absent)

faith, the stigma towards the person could arguably be greater than if attributed to the occurrence of temporary demonic possession.

Yet, etiological attributions go beyond merely situational or dispositional and involve a particular locus within an anthropological understanding of the person. From a biopsychosocial-spiritual perspective, views of suicide could also incorporate biological, psychological, social, or spiritual understandings, or various combinations of these. Although some might assume that religious views will be solely spiritual, biological attributions of mental illness are quite common among religious individuals (Mathews 2008). This suggests that religious views are not always reductionistic and can also be integrative of multiple levels of analysis (Myers 2010).

Key Takeaway for Researchers: Religious beliefs about the causes of suicidality may incorporate both spiritual and scientific understandings of suicide. Understanding the complex manifestations of etiological beliefs about suicide could allow for a more complete understanding of contributions to stigma. Interventions could be designed to increase scientific knowledge of suicide, decrease stigmatizing spiritual etiologies, or shift beliefs towards empowering spiritual etiologies.

### 6.1.2. Religious Consequences of Suicide

Another set of religious beliefs directly related to suicide are those addressing the consequences of suicide. In the action research framework, the consequences of suicide include religious beliefs about the various effects of suicide upon the decedent. These beliefs could possibly be grouped into three categories: punishment, deterioration and defilement, and glorification. Punishment beliefs would involve a Higher Power that enacts a punishment or withholds a benefit as a result of the death by suicide. One punishment belief involves the idea that those who die by suicide will go to Hell. This belief has been found to have a protective effect on suicidal ideation and attempt (Van den Brink et al. 2018). However, punishment beliefs may contribute to stigma towards PWSI and PWSB, especially if a suicide death is believed to lead to damnation, regardless of the person's behaviors or beliefs while living.

Deterioration and defilement beliefs would entail a state of being or an afterlife that falls significantly short of an ideal state as a result of the act of suicide, with the clarification that this outcome is not enacted by a Higher Power. These beliefs would include reincarnation to a lower state of existence, as seems to be present among Buddhist and Hindu belief systems. These beliefs could involve the perception that the person is temporarily or permanently made impure as a result of their suicide. One type of deterioration and defilement belief noted in the literature involves the person's soul being tainted by their death by suicide (Rottman et al. 2014).

The final set of religious perceptions could involve the belief that suicide will lead to a glorified or exalted state of being or afterlife. Such a belief has been attributed to the Vikings, who are thought to have held that those who die by suicide would be taken to Valhalla, though such depictions may have been historically inaccurate (Nagel 2018). Nevertheless, glorifying views of suicide do exist (Batterham et al. 2013) and could be influenced by religious beliefs.

Key Takeaway for Researchers: Researchers should investigate underlying patterns in belief systems about the consequences of suicidality to better understand the effects of these beliefs on stigmatizing or empowering responses. Interventions could be designed to alter the beliefs about these consequences by suggesting alternative consequences, emphasizing uncertainty about consequences, or decreasing the salience of these beliefs.

### 6.1.3. Religious Practices Related to Suicide

Although religious beliefs about suicide might be more readily assessed and studied, a robust understanding of the religious stigma of suicide requires further clarification on the role of religious practices in stigmatization. Whereas religious beliefs might be more relevant to stereotypes and prejudices, religious practices might be more relevant to

discrimination. Such practices would encompass the responses and rituals of those affected by suicide.

Religious Practices towards Suicide Decedents

The religious community typically has a proscribed response to the death of someone who had connections to the community (Hays and Hendrix 2008). This response can include funeral services, burial ceremonies, and bereavement practices, among others. Understanding these practices and how they might differ for those who die by suicide can lend insight into the stigma of suicide. These practices could involve how suicide is discussed at the funeral, if at all, specific burial practices or locations of burial, and bereavement practices employed by SLS. Deviations from typical practices could potentially indicate positive support for those grieving through the affirmation of the unique aspects of death by suicide but could also reflect and perpetuate various stigmatizations. Understanding these practices and how they are understood within religious communities would be useful for developing effective interventions that minimize stigma and empower those affected by suicide.

Religious Practices towards Suicide Loss Survivors

Those who are bereaved are often supported by religious communities (Hays and Hendrix 2008). However, religious communities may not show the same support towards SLS. The response by the religious community may reflect blaming or judgment for the suicide or other forms of stigma. In contrast, some religious communities provide more extensive support for those affected by suicide loss, which may empower this population in their grieving process and reduce the burden of stigma.

Religious Practices of and towards PWSI and PWSB

Finally, it is important to survey the various religious practices employed in response to personal experiences with or disclosure of SI or SB. This could involve various forms of religious coping, including the use of intercessory prayer, visitation, confession, and sharing of religious resources, such as Scripture passages. The religious community's use and recommendation of particular religious practices might be a form of support but might also be a form of discrimination. Similarly, PWSI and PWSB may utilize different forms of religious practices than those used with other health conditions, such as exorcism, and this may reflect coping strategies unique to the challenges of suicide but might also reflect self-stigma. Research on this topic could contrast religious practices towards PWSI or PWSB against those employed during somatic illness or social crises to better delineate these practices and their functions within the religious community.

Key Takeaway for Researchers: Religious practices might be an important determinant of stigmatizing experiences. Studies ought to investigate how religious practices and forms of support differ for populations impacted by suicidality.

*6.2. The Relationship of Broader Religious Worldview with Suicide Beliefs and Practices*

It is important to recognize that religious beliefs about suicide did not develop within a vacuum and are instead shaped by broader religious teachings within a particular cultural context. For instance, the Catechism of the Catholic Church reads, "Everyone is responsible for his life before God who has given it to him. It is God who remains the sovereign Master of life. We are obliged to accept life gratefully and preserve it for his honor and the salvation of our souls. We are stewards, not owners, of the life God has entrusted to us. It is not ours to dispose of" (Catholic Church 2019, para. 2280). This belief is grounded in theological convictions that have broad implications across a variety of domains but, nonetheless, this teaching also has clear inferences for beliefs about suicide. Religious teachings about suicide should thus be interpreted within the context of these broader belief systems.

6.2.1. Religion and Ethics

In regards to ethics, religious beliefs can include general positions indirectly related to suicide and specific positions on the moral nature of suicidal actions. It is critical for research to explore the full array of these ethical positions as they mediate the relationship between religion and suicide. Although the principle of compatibility (Ajzen 1988) would indicate that beliefs that are more specific to the behavior will have a stronger bearing on attitudes and intentions for engaging in a particular behavior, indirect beliefs may nevertheless play a role in shaping these beliefs in aggregate. For instance, affirmation of the value of life might not strongly predict attitudes against suicide but may be foundational for this belief.

Religious prohibitions against suicide, although similar in content, may be achieved through vastly different ethical systems reflecting religious perspectives. Religious ethics could be broadly classified as consequentialist or deontological. From a consequentialist perspective, such as utilitarianism, suicide can be judged according to its effects on all involved (Hooker 2016). Thus, suicidal ideations, attempts, and behaviors can be determined to be immoral based upon the consequent harmful effects on the community. These effects could involve merely pleasure and pain or could involve an understanding of the welfare of the community. From a deontological perspective, suicidal behaviors can be judged based upon the intentions of the actor, the action itself, or the conjoined intention-action (Alexander and Moore 2021). The notion that suicide taints the purity of the person's soul is based on deontological ethics (Rottman et al. 2014).

These ethical concerns could be investigated with empirical research to provide clarity on the associations with specific suicide beliefs and stigma. One empirical study investigated consequentialist and deontological ethics and found that judgments against suicide reflected purity concerns rather than beliefs about the harm to the community (Rottman et al. 2014). In that study, it was found that participants' belief that suicide corrupted the purity of the person's soul, along with disgust towards suicide, were associated with the belief in the moral wrongness of suicide. Research could investigate the degree to which religious traditions differ in the ethical underpinnings in their views of suicide.

Another possibility could be that religious ethical views and practices could inhibit other risk factors for suicide, such as substance use (Chitwood et al. 2008), impulsive behavior (Caribé et al. 2015), and hostility (Lutjen et al. 2012). It would be important to investigate how ethical beliefs related to these practices are related to suicide risk and stigma towards suicide.

At the same time, it would be important to investigate how broader religious beliefs and practices might promote suicidal thoughts. For instance, believing that one has committed an unforgivable sin has been found to be associated with suicidal thoughts (Exline et al. 2000). Such beliefs might reflect negative religious coping, or religious strain, where the person believes that God is punishing or has abandoned the individual (Pargament et al. 2011). The Hindu practices of *Sati*, where a widow jumps upon the funeral pyre of their deceased husband, and *Prayopavesha*, where a person starves themselves to death as a means of achieving enlightenment (Teo et al. 2021), might also promote suicidal tolerance. Within Buddhism, self-immolation is held up as an act of devotion that expresses gratitude to the Buddha, who made it possible to use substitutionary sacrifices to express devotion (Teo et al. 2021). Another example would be Islamic views of women as inferior, which could arguably cause suicide, often by self-immolation, to protest their oppression or abuse (Teo et al. 2021). Understanding the underlying religious, ethical frameworks that give rise to such thoughts would aid in equipping faith leaders and clinicians in knowing how to respond.

Key Takeaway for Researchers: Investigating religiously-based ethical systems could provide insights into suicide-specific beliefs and practices that may have implications for stigmatizing and empowering responses to suicide. The use of culturally-relevant suicide prevention methods based in these beliefs and practices can further assist researchers to design effective interventions.

### 6.2.2. Religion and Anthropology

Another broad aspect of religion that could have implications for the stigma of suicide are anthropological. Religions make various claims about the nature of personhood that might have consequences for views of suicide. For instance, Buddhism upholds that people are characterized as having *tanha*, or unenlightened worldly desire, which seems to shape views of suicide. Similarly, the belief that people are created in the image of God and that God has entrusted people with their lives seems to be critical to Catholic theology of suicide (Catholic Church 2019, para. 2260, 2280).

One aspect of anthropology that may play a critical role in suicide stigma is beliefs about the presence of a non-material essence (e.g., soul). One view of the person, shaped by Christian theology, was advanced by Descartes in the 16th century and involved seeing people as composed of a hydraulic body and a non-material mind that were distinct but interconnected (Robinson 2020). This dualistic view of personhood could be contrasted with monism, which sees the person as being of essentially a single substance. A dualistic perspective could potentially facilitate an either/or perspective that would attribute suicide to either the body, and thus one's biology, or the mind, and one's agency and choice (Forstmann and Burgmer 2018). Religiousness could influence the development of dualist or monist views of the person, or potentially other anthropological understandings. As an example, Jewish views of the person tend to be more holistic and less dualistic in viewing persons as a unified body and mind (Satlow 2015). These anthropological understandings may influence stigma towards suicide, as stigma has been argued to result from a belief that suicide will taint the person's soul (Rottman et al. 2014).

Although a comprehensive review of the relationship between anthropology and religion is beyond the scope of this article, it is important to acknowledge its importance in shaping views of suicide, such as typologies, etiological beliefs, and practices related to suicide. Investigating the relationship between such constructs might be helpful in potentially counteracting the religious stigma towards suicide in a manner that is religiously congruent.

Key Takeaway for Researchers: Religious beliefs about the nature of personhood may, directly and indirectly, affect stigma towards suicide. Studies should investigate these relationships to provide a robust understanding of suicide stigma.

### 6.2.3. Religion and Afterlife Beliefs

One of the major functions of religion is to provide an understanding of death, including life after death (Moreman 2017). Children develop these afterlife beliefs as young as five years old (Bering et al. 2005). The beliefs about the afterlife can vary tremendously between religions, with Islam and Christianity upholding beliefs in Heaven and Hell, Judaism focusing less on the afterlife, while Buddhism and Hinduism uphold belief in reincarnation (Gearing and Lizardi 2009; Nelson et al. 2012). Religion's role in facilitating certain afterlife beliefs ought to be studied with suicide stigma, given that belief in Hell has been found to be associated with decreased suicidal ideation and suicidal behavior (Van den Brink et al. 2018).

### 6.3. Effects of Religious Beliefs and Practices on Stigma towards Suicide

A key question is whether particular views and practices related to suicide, as well as broader religious and practices, lead directly to stigmatization of suicidality. In regards to religious views of suicide, it is critical to distinguish between beliefs about suicide and stigma towards the suicidal person. As noted above, Link and Phelan (2001) argued that stigma eventually results in discrimination towards the person. Religious beliefs about suicidal ideation or behaviors, including the perceived morality of these behaviors, do not require the person to be labeled, set apart, and discriminated against. Attributions of suicide can involve situational causes, temporary states, or deviations in overall character that may impede a global label from being applied to the person. In religious terminology, the "sin" does not necessitate applying a particular label to the "sinner." Religious frameworks can

aid in making this distinction, such as attributing suicide to systemic evil. Nevertheless, it is unclear if certain types of beliefs, such as condemnation to Hell for engaging in suicide, are so severe that they require a label be applied to the individual.

Another key aspect of the relationship between religion and the stigma of suicide is understanding the function of the religious proscriptions. Phelan et al. (2008) reviewed various models of stigma and argued that they cohere around three main functions of stigma: exploitation and domination, norm enforcement, and disease avoidance. To better understand how to address the stigma of suicide, researchers ought to investigate the degree to which these functions are at play in the religious stigma towards suicide. From a casual perspective, it would seem norm enforcement and disease avoidance could be at play. Suicide proscriptions could be thought to achieve norm enforcement through re-integrative shaming (Braithwaite 1989) with the assumption that suicidal behaviors are within a person's control. Alternatively, or perhaps additionally, religious stigma towards suicide could function as a means of disease avoidance. From this perspective, there may be evolutionary mechanisms at play that trigger a disgust response to suicide that functions to distance people from those who may have an infectious disease or harmful genetic mutation (Phelan et al. 2008).

In regards to disease avoidance, Rottman et al. (2014) have advanced an argument that the underlying root of the stigma of suicide is evolutionary, as opposed to cultural. In two studies, the belief in the wrongness of suicide was predicted by beliefs that suicide taints the purity of the soul, a belief that was closely associated with the emotional response of disgust. This was in contrast with homicide, which was predicted by beliefs about harm and was associated with the emotional response of anger. The authors argued that this response to suicide implied that rational appeals would be ineffective at decreasing the stigma of suicide because it is an evolutionarily determined process, unlike the harms of homicide, which can be altered with cultural appeals (e.g., beliefs that violence is justified; see also (Rottman and Kelemen 2014) for commentary on suicide terrorism). Their study also found that most people were unaware of the role of purity beliefs in predicting their moral judgments, as beliefs about harm were relatively stronger, and that the findings applied regardless of whether the individual was religious or not.

The argument that stigma towards suicide is evolutionarily fixed rather than culturally dictated would suggest that attempts to alter religious proscriptions against suicide would have little effect in reducing stigma. However, other research concluded that disgust is mediated by beliefs about harm (Schein et al. 2016). Moreover, this argument overlooks particular views of suicide across history that have glorified suicide, such as the Stoicism, Vikings, and Eskimo beliefs about violent deaths, the Japanese practice of *hari kiri*, or the Hindu practice of *sati*. Therefore, the conclusion of Rottman et al. (2014) that stigma towards suicide reflects evolutionarily ingrained disgust responses that will not be amenable to intervention may be premature. Nevertheless, addressing the underlying cognitive processes through which stigma towards suicide arises would be critical for understanding the relationship between religion and the stigma of suicide.

Key Takeaway for Researchers: Research is needed on the cognitive processes and community functions of religious beliefs and practices about suicide in regards to stigma, including potential salubrious outcomes.

### 6.4. Ecological, Cultural, and Multidisciplinary Perspectives of the Religious Stigma of Suicide

The ARF emphasizes that religious beliefs and practices, along with stigmatization, occur within broader ecological and cultural systems. Suicide research has often lacked in reporting on diversity data such as religious affiliation, ethnicity, and disability status (Cha et al. 2017). Although there has been a strong focus on individual psychological factors (Franklin et al. 2017), suicide research rarely accounts for ecological or sociological factors such as culture, religious identity, and involvement, or stigma in a generalizable manner (Franklin et al. 2017; Cha et al. 2017; Bowden et al. 2020). Furthermore, aside from studies focusing specifically on small minority groups (i.e., a small group of refugee

immigrants), religious and cultural beliefs, practices, involvement, behaviors, coping resources, and support systems are essentially entirely overlooked and rarely measured in suicide research (Franklin et al. 2017; Cha et al. 2017; Bowden et al. 2020) or even included in measures of suicide stigma (Ghasemi et al. 2015). Incorporating a more multidisciplinary perspective of suicide and stigma through the inclusion or consideration of theological, sociocultural, ethnoracial, philosophical, historical, political, and liberal arts perspectives in addition to a number of interdisciplinary clinical and research perspectives may function to more effectively integrate issues of intersectionality, culture, and religion in suicide research, prevention, intervention, postvention, and in the development of clinical practice guidelines.

### 6.4.1. Ecological Systems Theory

Bronfenbrenner's ecological systems theory (EST) proposes an individual exists and develops within a complex system of relationships and environments (Bronfenbrenner 1979). Applying EST in consideration of suicide allows researchers to account for how culture, religion, and the intersection of the two beget relationships and environments influencing the ways in which people conceptualize and understand issues of suicide, stigma, death, mental illness, help-seeking behavior, and a number of other factors associated with suicide. The relationship between these and other confounding variables, suicidality and stigma, is further complicated by the reality of both the intersectional nature of identity and by cultural and religious changes over time. More specifically, research on suicide and religious affiliation alone does not demonstrate a clear causal relationship due to the confounding nature of intersectional identity through factors such as ethnoracial and cultural identity (Maris 2019) and other variables such as religious activity, practices, and behavior.

### 6.4.2. Cultural and Diversity Considerations

As mentioned earlier, culture and religion are closely connected. The intersection of religion and culture has been discussed using the constructs of individualism and collectivism (Cohen et al. 2016). While societies vary in terms of how collectivist or individualist they are, it is important to note that these constructs are multidimensional (Triandis 1996; Triandis and Gelfand 1998). Hui and Triandis (1986) defined the constructs, in part, based on the goals of an individual. They stated that individualism is a primary focus on one's own goals and well-being, while collectivism is a primary focus on the goals and well-being of the group or unit over the individual. Several studies have analyzed the prevalence of individualism and collectivism in different societies. In general, the United States and countries in Western Europe tend to evidence strong individualism while East Asian countries evidence strong collectivism (Fijeman et al. 1996; Kitayama et al. 2010; Morling and Lamoreaux 2008).

Memhet Eskin (2013) sought to identify how an individualist perspective would impact one's suicidal ideation and behavior as opposed to a collectivist perspective in the country of Turkey. Eskin's research identified higher rates of suicidal ideation and behaviors in Turks who were more individualist than those who were more collectivist. Eskin noted that other factors may be associated with those who identified as more individualistic, namely dispositional causal attribution, as opposed to more situational causal attribution in collectivist-oriented individuals. Cultural variables such as these may also reflect the religious influence and, thus, a more robust understanding would investigate the intersection of religion and culture in suicidal ideation and behaviors, as well as stigmatization of PWSI and PWSB.

A related cultural dynamic is honor. Crowder and Kemmelmeier (2017) define honor culture in the US as the connection between one's self-worth and their public reputation. The researchers noted that individuals in the southern part of the US as well as the western part of the US identified more with honor culture. In light of the established research that shows that older men of European ancestry are at the greatest risk of suicide, Crowder and

Kemmelmeier's study identified a link between older men of European ancestry and honor culture. They posited that older men of European ancestry may find it more challenging to uphold their public reputation, thereby putting them at greater risk of suicide. This study and the study by Eskin are good examples of how future research would benefit from the nuances of culture and its impact on suicide.

Another important set of factors that intersect with religion and suicide is sex, gender identity, and sexual orientation. Sex differences as it relates to suicide have been examined in the research, with males having higher rates of suicide than females in nearly every country (Hawton 2000). Moreover, sexual and gender identity presents an important concern in suicide prevention and intervention efforts, namely, individuals in the LGBTQIA+ population are found to have higher rates of suicide (Blosnich et al. 2013; Plöderl et al. 2013). This concern is further compounded when considering intersectionality due to the role of religious ideology in the development and perpetuation of beliefs around sexual and gender identity and expression. Religiousness has been found to be associated with differences in the prevalence of suicide between males and females (Alothman and Fogarty 2020) and is also implicated as a risk factor for suicidal ideation and suicide attempt among sexual minorities (Lytle et al. 2018). Addressing the intersection of religion, sex, gender, and sexual orientation with suicide, such as beliefs about the relationship between sex and gender, normative gender roles, and proscriptions towards particular sexual behaviors, would yield a more complete understanding of these dynamics.

Another issue of note is the role of ethnoracial identity in relation to religious beliefs and practices around issues of suicide. For example, it is generally understood denominational differences exist in suicide mortality rates; however, sometimes those denominational differences are not present in certain ethnoracial groups within the lower-risk denominations (Maris 2019). Some of these differences may be due to factors such as political influence in post-Soviet nations or hardship resulting from war or changes in national economies (Lester 1999); thus, the incorporation of a multidisciplinary perspective that considers intersectionality may not only be helpful but integral in understanding the way in which different aspects of ecological systems affect suicidal ideation, intention, and behavior.

6.4.3. Multidisciplinary Perspectives

When considering how research on the relationship between religion and the stigma of suicide could advance, it is also important for scholars to recognize the value of research from other disciplines. Although the model has placed a social psychological perspective of stigma as the central framework, self-report measures commonly utilized in this discipline may not be as effective at capturing the cultural beliefs that actually influence discriminatory behavior (Pescosolido 2013). Sociological research can help capture differences in stigma at the national and cultural levels, as well as understanding how power structures perpetuate this stigma.

Moreover, the psychological and sociological findings on this topic ought to be complemented with philosophical and theological considerations, as well as the broader liberal arts and humanities. Historical accounts of the relationship between suicide and religion could provide insight into contemporary problems. Communication studies could investigate how rhetoric related to suicide and religion might affect cognitive and emotional responses. Scholars from countless other fields could contribute to this endeavor. Multidisciplinary empirical and theoretical studies could provide insights for conceptualizing and intervening religious stigma towards suicide.

Another perspective could be philosophical and theological anthropology. This could accord insight into the relationship between a person's behavior and their nature as a person. The knowledge about religious views of personhood might provide insight into how various religion views could affect how a person engages in suicide behavior and what the action could mean about their identity as a whole. Conversely, philosophical and theological positions might be predicated on false notions and thus be better informed

by contemporary research. For instance, some theologians and faith leaders assume that prohibitions against suicide are effective deterrents against suicide, even though the individual may personally disagree with this position. If studies were to clearly demonstrate that this is not the case, that might lead to significant shifts in how theologians and faith leaders address suicide in academia or church settings.

Key Takeaway for Researchers: Research ought to consider the role of religious beliefs and practices about suicide within ecological systems and with consideration of the intersection with diverse identities and expressions. Multidisciplinary research is encouraged to broaden the understanding and impact of research on religious beliefs and practices about suicide.

### 6.5. Research Design and Analytic Approaches

Advances in technology and statistical techniques allow for the measurement of more complex relationships between and among variables, such as conditional indirect effects, likely through the development of suicide risk algorithms and the application of machine learning. This may yield better predictability and generalizability compared to the past 50 years of suicide research, which has focused mainly on risk factors and yielded few to no predictive utility (Franklin et al. 2017). It is possible the inclusion of systems-level factors in suicide research and the application of more advanced statistical techniques may aid in the development of predictive risk algorithms specific to certain groups or populations that demonstrate greater predictive utility than the general application of current theories. For example, a network analysis of risk factors demonstrated indirect relationships between factors of two main theories of suicide and current suicidal ideation (DeBeurs et al. 2019). Further research could consider findings of this research in the development of risk algorithms. Additionally, the inclusion of systems-level diversity factors may allow researchers to better understand the ways in which factors in individuals' ecological systems interact to affect both suicide risk and the relationship with stigma.

Key Takeaway for Researchers: Researchers ought to use research designs and data analytic techniques appropriate to the research question, including more sophisticated methods if they might bring about useful insights relevant to the topic.

### 6.6. The Intersection of Stigma Reduction and Suicide Prevention

Research agendas related to religion and suicide stigma ought to seek knowledge that will support stigma reduction and suicide prevention. However, there exists a perception that religious stigma about suicide might actually protect against suicide (Mason et al. 2021). Historical accounts of the theological views of suicide argue that the beliefs may have constrained many from taking their own lives by suicide (Barry 1995). In support, suicide tolerance has been found to be associated with religious beliefs and has a protective effect on suicide (Neeleman et al. 1997). Some argue that therapists working with suicidal clients should not challenge religious views of suicide that are punitive or condemning, arguing that these views are effective in discouraging suicidal behaviors (Page 2018). Indeed, one review of studies on moral objections to suicide concluded that these beliefs were associated with less suicidal ideation and suicidal behaviors, although the authors did not make causal claims (Van den Brink et al. 2018). This is bolstered by the clinical observation that many people report that fear of being condemned to Hell is a deterrent to taking suicidal action.

However, the idea that religious stigmas towards suicide actually prevent suicide attempts and death by suicide contradicts other findings. There is growing literature that stigma actually increases the risk of suicide. Some experts have pointed out that the consequences of stigma, such as social isolation, unemployment, and hopelessness, are risk factors for suicide (Rüsch et al. 2014). Others have argued that the relationship between suicide and stigma is reciprocal, with suicidal thoughts and behaviors being stigmatized and, in turn, the stigma of having a mental illness and suicidal behaviors leading to increased suicidal thoughts (Carpiniello and Pinna 2017). The stigma can be a

result of having a mental illness, with some attempting suicide as an attempt to escape from the perceived stigma (Eagles et al. 2003).

The few studies that have been conducted on the relationship between stigma and suicide have indicated that stigma contributes to suicide risk. In one study, a path analysis found that self-stigma and anticipated public stigma each independently contributed to suicidal ideation (Oexle et al. 2018). Consistent with this notion is that national rates of suicide have been found to be lower in countries with greater social acceptance of a person with a significant mental health problem (Schomerus et al. 2015). Similarly, a longitudinal study found that self-stigma was associated with suicidal ideation at one-year and two-year follow-ups. Lending further weight to the positive linkage between suicide and stigma is the finding that countries with increased stigma towards mental disorders had higher rates of suicide (Schomerus et al. 2015).

Yet, it is currently unclear whether religious stigma towards suicide functions to exacerbate suicide risk similar to other forms of stigma. The conclusion mentioned above that moral objections to suicide were associated with decreased suicidal ideation and suicidal behaviors might suggest this to be the case (Van den Brink et al. 2018). However, it is important to underscore that the authors noted that the conclusions were based on a small set of cross-sectional studies of "fair to poor" quality and thus refrained from conclusions of causality. It is plausible that suicidal ideation and behavior may have an inhibiting effect on moral objections to suicide, due to cognitive dissonance or some other mechanism, such that the direction of this relationship is reversed.

Given the uncertainty of the effects of religious stigma on suicide risk, investigating the possibility that religious stigma might exacerbate risk is critical. Arguably, religious stigma could contribute to anticipated stigma and self-stigma for those with a history of suicidal ideation or behavior, thereby increasing hopelessness or feelings of entrapment and consequently increasing suicidal ideation intensity. Another possibility is that various types of religious stigma may have counteracting effects. For instance, religious stigma about afterlife consequences may indeed inhibit suicidality but religious stigma about the causes of suicide, such as "weak faith," may still be associated with increased risk of suicide. Teasing these effects apart could potentially differentiate the most pernicious types of stigma to address in interventions from those that should be left intact. Finally, the intersection of religious stigma with religious support should be examined. Religious stigma towards suicide may differ in its effects on suicide risk because it may be accompanied by social support. A study found that a sense of belonging moderated the relationship between self-stigma and suicidal ideation (Wastler et al. 2020). The religious stigma of suicide is often experienced within a religious context that can simultaneously provide social support that can inhibit suicide risk (Nelson et al. 2012).

The role of religious stigma in preventing or promoting suicide is a critical area within this ARF that requires a robust body of research. Although cross-sectional studies will continue to have an important role, researchers must consider utilizing longitudinal, proscriptive designs to examine how religious stigma impacts the risk of suicide over time (Van den Brink et al. 2018). Qualitative studies would also be valuable to explore novel experiences that might shed light on these relationships. Moreover, studies on suicide decedents, such as psychological autopsies, might offer clues for preventing further suicides. Studies could investigate the relationship between religious suicide stigma and suicide rates across regions or countries, similar to studies examining stigma generally (Schomerus et al. 2015). Together, these areas of research could provide an understanding of each individual's experience of their own religion (Lawrence et al. 2016), which can provide a comprehensive understanding for efforts to reduce stigma.

Key Takeaway for Researchers: In order to clarify guidelines for clinicians working with religious clients experiencing suicidal thoughts or behaviors, studies should investigate the relationship between religious suicide stigma and relevant outcomes, including suicidal ideation, suicide attempt, and death by suicide. When researching this topic,

study methodology should be selected to build a robust literature relevant to efforts at suicide prevention.

*6.7. Identifying Religious Resources for Reducing Religious Suicide Stigma*

The final component of the ARF is identifying religious resources and effective interventions for reducing the stigma of suicide. It is essential that researchers perceive religion not as an obstacle to suicide prevention but rather as a rich and complex resource that ought to be engaged collaboratively. For many individuals, religion is the most important aspect of their lives and may even be the fundamental source that makes their life worth living. Thus, viewing religion merely as a source of stigma towards suicide would be a considerable misunderstanding of the lived experiences of religious individuals. Those interested in reducing religious stigma towards suicide must be critically engaged in understanding these beliefs and their role in helping individuals and communities cope with difficult life circumstances, including suffering.

To pursue the reduction of the religious stigma of suicide, it is important to identify beliefs and practices involved in the stigmatization or empowerment of PWSI, PWSB, or SLS, that are malleable in religiously and culturally appropriate ways. Oftentimes, religions include various teachings that can have differential impacts on suicide stigma. For instance, Christianity teaches about the love of neighbor, which could reasonably decrease suicide stigma, and also punishment for one's wrong actions, which could increase suicide stigma. Identifying these beliefs and their divergent impact on religious stigma towards suicide can allow for emphasizing or de-emphasizing beliefs based upon their impact on people impacted by suicidality and suicide loss. When identifying these malleable beliefs, researchers should involve religious stakeholders, such as faith leaders and clergy, to ensure that interventions do not inflict unintended religious or cultural damage within a particular faith community.

Researchers ought to also investigate novel interventions for reducing stigmatization and increasing empowerment related to suicidality and suicide loss. Clinical interventions conducted by licensed mental health professionals comprise a significant and critical aspect of the response to suicide stigma. These interventions would be particularly appropriate for addressing personal stigma experienced by PWSI and PWSB and could be incorporated into evidence-based interventions for suicidality, such as Collaborative Assessment and Management of Suicidality (Jobes 2012), Cognitive Behavioral Therapy for Suicide Prevention (Stanley et al. 2009), or Dialectical Behavior Therapy (Linehan 2020).

However, these clinical interventions would likely be best complemented by other interventions targeted toward the general public, as personal stigma is the result of public stigma (Sheehan et al. 2018; Fox et al. 2018). Focusing merely on clinical interventions is shortsighted and neglects to effectively account for intersectionality, and the wealth and depth of cultural capital people can draw upon during times of hardship and distress. The narrow focus on individual risk factors has proven ineffective at suicide prediction (Franklin et al. 2017) and considering a broader perspective may prove fruitful in suicide prediction, prevention, intervention, and postvention.

The United States Substance Abuse and Mental Health Services Administration (SAMHSA) has published recommendations for culturally competent suicide prevention that seek to integrate heritage culture in prevention efforts. In addition, it even suggests the potential benefit of providing services and support in non-traditional systems and milieus instead of only those associated with traditional mental health treatment (Substance Abuse and Mental Health Services Administration SAMHSA). Some community-level interventions exist for disorders associated with high rates of suicide, such as Family Connections™, which is a peer-led program directed toward the loved ones of individuals with borderline personality disorder (Hoffman et al. 2005). It is possible an adaptation of this program to broader issues of suicide stigma may prove an effective way to provide support to PWSI, PWSB, their families, and their religious and cultural communities.

Equipping faith leaders with competencies to engage with suicidality is integral in suicide prevention (Mason et al. 2021; National Action Alliance for Suicide Prevention: Faith Communities Task Force 2019). Although the identified clergy competencies have included theological reflection on suicide, specific guidance on particular theological concepts related to suicide stigma within their religious tradition might be worthwhile. For instance, theological reflection could involve guided processes to identify suicide conceptualizations, beliefs about suicide causes, beliefs about suicide consequences, and other beliefs more distal but still relevant to suicide, such as ethics, anthropology, and afterlife beliefs.

Furthermore, it may also be beneficial to develop similar competencies for clinicians in integrating culture and religion into the support system of PWSI, PWSB, and SLS in the engagement of discussion around religious and cultural issues of suicide (e.g., fear of condemnation to hell). Providing guidelines for therapists to constructively and effectively engage these discussions to lessen both stigma and suicide risk, while respecting the client's religious diversity, would likely be valuable.

Key Takeaway for Researchers: Evidence-based interventions should be developed to target religious beliefs and practices that may be involved in both suicide stigma and suicide prevention. Interventions should be developed to be appropriate to the cultural and religious context.

## 7. Clinical Case Study and Action Research Framework Application

To elucidate the action research framework (see Figure 2), a hypothetical client's recent suicide attempt is provided below. Each part of the action research framework, including religious-cultural worldviews, religious-cultural suicide beliefs and practices, and responses to suicidality are explained through the case study of Fernando.

Client/patient: Fernando is a 23-year-old Latino male from Texas who works as a solar panel installer and identifies as Catholic but attends Mass irregularly. Fernando recently experienced a suicide attempt that resulted in medical attention at an emergency department followed by a transfer to a psychiatric hospital.

### 7.1. Family History

Fernando's father and mother married after becoming pregnant with Fernando but divorced when he was four years old. Fernando's father had a history of using methamphetamine and spent several years in jail for various theft- and drug-related charges. Fernando has little contact with his father and his father's family. Fernando's mother had a stable job throughout his childhood but lost legal custody after having two successive relationships where domestic violence and physical abuse of Fernando occurred. Fernando's maternal grandmother obtained legal custody of Fernando at age 11, though his mother continued to have a role in his life as they all lived with his grandmother. Fernando's maternal grandmother immigrated from the United States shortly after her husband, his grandfather, died by suicide while in El Salvador, leaving her and their two children, including Fernando's mother, as survivors.

### 7.2. Stigmatizing Views of Suicide

Fernando's view of suicide was primarily shaped by his grandmother who spoke openly and disparagingly of her deceased husband's suicide. She would speak of him as being "a worthless man who was a failure in everything he did." She would say that he was "selfish" for leaving her with two children and few resources. She claimed that those who killed themselves were terrible people and "deserve the punishment they receive." Fernando believed that suicide was "an act of cowardice" that meant "you are turning your back on your family." Fernando's views had fomented resentment towards his grandfather and fear about whether he "inherited" the problem of suicidality.

### 7.3. Religious Conceptualization of Suicide

Fernando viewed suicide as a "choice" to turn away from the obligations God has given a person to their family. His conceptualization emphasized suicide as a "decision" and "a failure to show responsibility." He viewed suicide as an unforgivable sin. He believed that thinking of suicide was forgivable but felt that a pattern of these thoughts reflected deep sinfulness in the person. Although he believed that some suicidal thoughts could be justified, such as when a close family member dies, he did not believe that his circumstances justified his thoughts of suicide. Regarding his suicide attempt, Fernando believed that his actions revealed a complete lack of faith in God. He felt deeply troubled that his thoughts about suicide would bring shame to his family, particularly his grandmother, given that he held his grandmother in high regard for her devotion to God.

### 7.4. Religious Consequences of Suicide

Fernando believed that God punished those who died by suicide to Hell because their souls would become defiled. He was unaware of the teachings in the Catechism that affirmed that psychological factors could lessen this responsibility. Fernando believed that thinking of suicide resulted in a disconnection with God, who was deeply disappointed when someone questioned the life that God had given. Fernando also believed that partaking in the sacraments while thinking of suicide would deeply offend God.

### 7.5. Suicide-Related Practices and Coping

When Fernando began experiencing thoughts of suicide, he felt deep shame and hid the thoughts from others. He avoided going to church for fear of offending God or worsening the shame he felt. Fernando's avoidance of Mass led him to feel more distressed as he believed church was the best way to "right his relationship" with God. His prayers expressed a deep desire for God to relieve these suicidal thoughts but he simultaneously felt conflicted that he did not deserve God's help, as he felt cursed by a legacy of the sin of suicide. At times he wondered if God wished for him to kill himself as though this would be the only way to demonstrate remorse. This thought led him to feel trapped and desperate and was the belief that preceded his suicide attempt.

### 7.6. Religiously Appropriate Interventions

A priest visited Fernando at the hospital and taught him about the Catholic Catechism's position on suicide and pointed out that his beliefs were more stringent than the official position. This priest normalized his thoughts of suicide using examples from Scripture and of the saints and reassured him that God welcomed him back into a life of faith through the sacraments. Taking the Eucharist was a particularly meaningful moment for Fernando, who felt a deep sense of relief when the priest offered him this opportunity. The priest provided psychoeducation to Fernando on the causes of suicide, including that many thinking of suicide perceive themselves to be a burden upon others, to counter the argument that suicide is an inherently selfish act.

## 8. Limitations and Considerations

The ARF provides a foundation for research on religious stigma towards suicide, but certain limitations must be considered. First of all, the multidimensional nature of stigma means that outcomes are likely to be complex and potentially contradictory. For instance, the possibility that certain beliefs about suicide may lead to increased blaming of a person but also greater optimism about the potential for recovery. In this case, whether the religious beliefs ought to be considered stigmatizing or empowering, or perhaps neither, may not be self-evident. Researchers will need to strive to seek outcomes that are religiously and culturally appropriate, rather than making assumptions about ideal outcomes of interventions.

Another limitation of the ARF is that it could potentially identify religious expressions without contextualizing these beliefs within the broader religious system. Identifying

specific beliefs about suicide and relating these beliefs to stigma might fail to understand the full function of these beliefs within the faith community or to the religious individual. Researchers utilizing the ARF ought to make efforts to contextualize beliefs within a historical, theological, psychological, and cultural context.

A final limitation is that, although the ARF was developed to be inclusive of various religious systems of belief and practice, the research team did not include representatives of different faiths. The research team was composed of those who identified as Christian, albeit with exposure to various sectarian traditions within that faith, including Protestant, evangelical, Catholic, and Orthodox. This shared identity may have resulted in a framework that over-emphasizes Christian systems of thought or practice and may not be as useful for other religious traditions. The research team acknowledges this limitation and would be open to adaptations of this framework to better include other religious traditions.

## 9. Conclusions

As noted at the onset of this article, this action research framework was primarily designed to aid researchers to develop strategies to reduce the religious stigma of suicide while simultaneously preventing suicide. The ARF provided a model of the religious stigma of suicide (see Figure 1) that was embedded within specific and broad religious beliefs and practices relevant to suicide (see Figure 2). By providing a clear layout for studying religious stigma, research can advance more systematically. Moreover, the ARF emphasizes the simultaneous goals of stigma reduction/empowerment and suicide prevention so that the well-being of PWSI, PWSB, and SLS could be supported. Researchers can utilize this framework for reviewing past research, designing research studies, interpreting findings, and proposing new intervention strategies to help these populations.

In addition, the ARF might be helpful for clergy and faith leaders, as well as mental health professionals working with religious individuals, to understand and perhaps curtail religious stigma towards suicide. It is important for clergy to engage in theological reflection in regards to their position on suicide (Mason et al. 2021; National Action Alliance for Suicide Prevention: Faith Communities Task Force 2019). The ARF suggests that clergy ought to reflect specifically and comprehensively on their conceptualizations of suicide, beliefs about the causes of suicide, and beliefs about the consequences of suicide. Moreover, the ARF aids in the assessment of personal stigma towards suicide that might be helpful for those intervening to reduce stigma and decrease the risk of suicide. However, the ARF should be seen as only a framework, rather than a direct guide for reflection, intervention, or assessment.

Although there is significant literature on religion and suicide (Gearing and Alonzo 2018), the intersection of religion, suicide, and stigma has been overlooked (Moksony and Hegedűs 2021). Given the vast numbers of religiously affiliated people impacted by suicidal ideation, suicidal behaviors, and suicide loss, along with the additional burden of stigma, comprehensive responses are warranted. This ARF will hopefully spur research on this topic as knowledge on this topic is expected to be critical for leveraging religious resources to protect against suicide, overcome suicide stigma, and promote well-being.

**Author Contributions:** Conceptualization, C.S.L., C.A.L., I.M., and S.M.G.; writing—original draft preparation, C.S.L., C.A.L., I.M., and S.M.G.; writing—review and editing, C.A.L., S.M.G., I.M.; visualization, C.S.L. All authors have read and agreed to the published version of the manuscript.

**Funding:** This research received no external funding.

**Institutional Review Board Statement:** Not applicable.

**Informed Consent Statement:** Not applicable.

**Conflicts of Interest:** The authors declare no conflict of interest.

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
