# Peer review of "An Action Research Framework for Religion and the Stigma of Suicide"

_religions, doi:10.3390/rel12100802_

Round 1

Reviewer 1 Report

The manuscript is an original consideration to guide a strategy within the quadrangle of suicide/religion/society/mental health. The suicide issue is of the highest clinical and social relevance. The material is likely to inspire the reduction of social-religious stigma directed at the suicidal. The construction follows an academic model and is based on solid literature.
The following considerations aim to adapt the manuscript to the editorial standards of Religions:

A. The first subheading (line 29) would be better replaced by "Introduction (or Background): Religions and Suicide," or something like.

B. In the "Scope of the Paper" section, it is necessary to describe where it came from and how the framework was developed. Is it the personal-professional experience of one or more authors? Is it the result of rounds of technical discussions between experts? A detailed paragraph dedicated to this would validate the whole reasoning.

C. Review some details from the References. Example: "Larkin, H.M. (2020). Religion, well-being, and stigma (Doctoral dissertation)." From which department of which university? Completed or under evaluation?

D. It would be pretty helpful to add to the manuscript:

  • a short paragraph (3-5 lines) at the end of each section about the main data of that passage. I see these data as the "bricks" for constructing the following reasonings, as shown in Figures 1 and 2. Otherwise, it isn't easy to retrieve the data from the text.
  • a case or situation (even invented) to illustrate the application by researchers (or clinicians or clergy) of the recommendations presented. Without this, the text could be a dry theoretical fabric.
  • before the Conclusion, a section dedicated to the Limitations of the study at hand, including what future studies should address. Just as an example (it doesn't have to be this): would this framework work equally in the western and eastern hemispheres?

Author Response

The authors would like to thank Reviewer 1 for the helpful feedback that contributed significantly to the quality of the paper. In response to comment A, the authors have made the suggested heading change. In response to comment B, the authors have added a description of the origination of the framework in lines 96-105. In response to comment C, the authors have reviewed the references and updated according to APA style. Comment D had three components and the authors have incorporated each of these suggestions. In response to the request for “bricks” to aid in constructing the reasoning, a brief summary was provided at the end of each section where the ARF is described with the subheading “Key Takeaway for Researchers.” A case study was also added, with incorporated commentary on relevance for the ARF, to keep the paper from being “dry theoretical fabric.” Finally, a limitations section was added prior to the conclusion, as requested by the reviewer.  

Reviewer 2 Report

This paper covers a very important topic. The authors successfully discuss a very wide array of social views on suicide and the stigma associated with it – as well as, accordingly, impacts on the bereavement process of those who lose family members and friends in this way. The diagram is also useful in visualizing the different sources. The issues of prejudice, stereotyping, and discrimination are nicely discussed, as is the potential for perception of a stigma that may or may not exist in society in a particular case.

It is also important that they successfully treated the issue of impact of stigma on those with suicidal ideation, e.g., social distancing, which can actually increase likelihood of suicide and decrease the potential for getting help to prevent it. Apparently this is been a major problem during the lockdowns and other measures during the pandemic. If correct, then lives were traded for lives. I mention that as another example of the problem the authors discuss, just in a different situation.

The authors also nicely provide a very broad and robust description of a wide variety of ways in which religion and religious organisations can help reduce suicide. This to me is exceptionally important given the dramatic secularization of a large portion of the world today. It is a resource for reducing suicide that appears to be underutilized. The authors propose what could be termed a "treatment plan," both individually and for society.

Author Response

The authors would like to thank Reviewer 2 for their helpful feedback. Reviewer 2 did not list any required changes to address. We would like to express our gratitude to Reviewer 2 for their encouraging comments and feedback.